# Analysis of Genetic Relatedness between Gastric and Oral *Helicobacter pylori* in Patients with Early Gastric Cancer Using Multilocus Sequence Typing

**DOI:** 10.3390/ijms24032211

**Published:** 2023-01-22

**Authors:** Ryoko Nagata, Hiroki Sato, Shoji Takenaka, Junji Yokoyama, Shuji Terai, Hitomi Mimuro, Yuichiro Noiri

**Affiliations:** 1Division of Cariology, Operative Dentistry and Endodontics, Faculty of Dentistry, Graduate School of Medical and Dental Sciences, Niigata University, Niigata 951-8514, Japan; 2Division of Gastroenterology and Hepatology, Faculty of Dentistry, Graduate School of Medical and Dental Sciences, Niigata University, Niigata 951-8510, Japan; 3Division of Genome-Wide Infectious Diseases, Research Center for GLOBAL and LOCAL Infectious Disease, Oita University, Oita 870-1192, Japan

**Keywords:** *Helicobacter pylori*, genotypes, allele sequences, gastric mucosa, multilocus sequence typing

## Abstract

The oral cavity is the second most colonized site of *Helicobacter pylori* after the stomach. This study aimed to compare the genetic relatedness between gastric and oral *H. pylori* in Japanese patients with early gastric cancer through multilocus sequence typing (MLST) analysis using eight housekeeping genes. Gastric biopsy specimens and oral samples were collected from 21 patients with a fecal antigen test positive for *H. pylori*. The number of *H. pylori* allelic profiles ranged from zero to eight since the yield of DNA was small even when the nested PCR was performed. MLST analysis revealed that only one patient had a matching oral and gastric *H. pylori* genotype, suggesting that different genotypes of *H. pylori* inhabit the oral cavity and gastric mucosa. The phylogenetic analysis showed that oral *H. pylori* in six patients was similar to gastric *H. pylori*, implying that the two strains are related but not of the same origin, and those strains may be infected on separate occasions. It is necessary to establish a culture method for oral *H. pylori* to elucidate whether the oral cavity acts as the source of gastric infection, as our analysis was based on a limited number of allele sequences.

## 1. Introduction

*Helicobacter pylori* is a major causative pathogen of gastritis, peptic ulcers, and gastric carcinoma [1,2,3,4]. Transmission of *H. pylori* is thought to occur mainly via the oral cavity during childhood [5,6]. *H. pylori* persists throughout the life of the patient without specific therapy, and most people infected with *H. pylori* usually remain asymptomatic. However, 30% of individuals may develop mild to severe upper gastrointestinal disease [6]. The prevalence of *H. pylori* infections among Japanese children and adolescents has been markedly decreasing with socioeconomic development. A meta-regression analysis of *H. pylori* infections in Japan from 1908 to 2003 revealed that the predicted prevalence in persons born in 1990 and 2000 was only 15.6% and 6.6%, respectively [7]. Thus, it is thought that the incidence of gastric cancer will continue to decrease over time [8].

As the oral cavity is the inlet port, researchers considered that it may also be the secondary habitat of *H. pylori* following colonization of the stomach. *H. pylori* grows spirally in the stomach, which is a microaerobic environment. In contrast, the bacteria are considered viable but non-culturable in the human oral cavity, as they do not appear to grow under aerobic or anaerobic conditions [9]. Therefore, nested polymerase chain reaction (PCR) is typically used to confirm their presence in oral specimens [10,11,12,13]. The specific DNA of *H. pylori* has been detected in a broad range of oral specimens, including saliva [13,14,15,16,17], supra- or sub-gingival biofilm [13,15,16,17,18], dentin caries [18], dental pulp [17,19], infected root canal [18], and the coating of the tongue [13,15]. We have reported that one-fifth of Japanese young adults have oral *H. pylori* DNA [13].

It has not been fully investigated whether *H. pylori* exists alive in the oral cavity and serves as a supply source for the stomach. The speculation that the oral cavity could also be a potential reservoir for *H. pylori* is based on epidemiological studies demonstrating a correlation between the presence of oral and gastric *H. pylori* [18,20,21]. However, so far, no evidence of a genotypic correlation between strains in the gastric mucosa and the oral cavity exists. Moreover, although there were some studies demonstrating that *H. pylori* in the oral cavity affected the outcome of eradication therapy [11] and that adjunctive periodontal therapy could enhance the efficiency of *H. pylori* treatment [22,23], there is still no evidence that the recurrence after eradication therapy of gastric *H. pylori* is caused by oral *H. pylori*. In their systematic review and meta-analysis, López-Valverde et al. reported that there is no clear evidence that *H. pylori* present in oral bacterial biofilm cause gastric infections and vice versa [24].

Since there are few studies comparing genotype differences in multiple genes, we aimed to investigate the genetic relatedness between gastric and oral *H. pylori* in patients with *H. pylori* infection in the stomach. To overcome the difficulty of the analysis, we applied a multilocus sequence typing (MLST) technique. MLST, which consists of a combination of partial nucleotide sequences of several housekeeping genes, is valid to address the intraspecific phylogenetic structure between different strains of the bacteria [25,26,27].

## 2. Results

### 2.1. Characteristics of Patients

The characteristics of the participants are summarized in Table 1. We recruited 21 patients (seven women and 14 men; mean age: 72.8 years, range: 59 to 86 years). Of the 21 participants, 20 had adenocarcinoma in the stomach, and one had squamous cell carcinoma and adenocarcinoma in the esophagus and stomach. The No. of remaining teeth and DMFT ranged from 5 to 30 and from 5 to 28, respectively. The average number of oral cleanings per day was 1.8, and 47.6% (10/21) of the patients used oral cleaning aids such as dental floss and interdental brush. Further, 19.0% (4/21) of the participants were smokers.

### 2.2. Identification of H. pylori in Stomach Tissue

The presence of *H. pylori* in gastric tissue was confirmed by both histopathology and PCR amplification of the 16s rRNA gene from the gastric biopsy. Hematoxylin and eosin (H-E) and Giemsa staining of the tissues showed spiral rods that penetrated between the gastric mucosa and mucus-producing cells (Figure 1 and Appendix A). This is consistent with what has been reported as a characteristic of *H. pylori* [28,29]. *H. pylori* DNA was detected in all participants by single-step PCR from an isolate on the agar plate (Appendix A).

### 2.3. Nested PCR and Allele Analysis

*H. pylori* DNA was detected in at least one site of the oral cavity in all patients using nested PCR (Table 2). The upper incisors had the highest abundance of the organism (71.4%, *p* < 0.05). Further, *H. pylori* DNA was detected in the biopsy specimens of all patients.

Table 3 compares the *H. pylori* allelic profiles obtained from oral and gastric samples based on eight housekeeping genes. The number of *H. pylori* allelic profiles ranged from zero to eight since the yield of DNA was small even when the nested PCR was performed. The alleles of eight loci from both collection sites were determined from only one patient (patient no. 3), and two out of eight alleles matched between oral and gastric samples. Moreover, for one sample set (patient no. 4), no amplification was observed for any of the eight housekeeping genes, and therefore, no allele analysis was performed for the sample set. For the rest, allele analysis was performed on the obtained sequences to determine the allele number, of which some were novel and, therefore, unclassified.

First, we evaluated the validity of the sequence obtained from the sample by creating a phylogenic tree using the allele type sequence here. Appendix A shows the phylogenetic analysis of the oral and gastric *H. pylori* based on the allelic profiles of the seven housekeeping genes. In addition, a total of 159 isolates from Asia, Oceania, Europe, North America, South America, and Africa were included in the phylogentic analysis. In this analysis, Asia was further categorized as East Asia, Southeast Asia, and South Asia. The *H. pylori* allelic profiles from the oral and gastric samples in this study mainly clustered with the Asian isolates. Partial sequences of the seven genes per patient are presented in the Appendix A.

### 2.4. MLST

The genetic relatedness between oral and gastric *H. pylori* was investigated using MLST. DNA sequences of up to seven of the housekeeping genes were used for MLST to obtain sequence types. Although the vacuolating cytotoxin A (*vacA*) gene was able to generate an allele number, it could not be used for MLST analysis on the PubMLST database (http://pubmlst.org/helicobacter/. Accessed on 17 December 2022) [30,31]. Table 4 summarizes the MLST profiles of the sample sets. Of the genotypes obtained from the oral cavity and stomach, only one sample set (patient no. 13) had a matching sequence type. The genotypes of the other 20 sample sets were various, and the gastric and oral *H. pylori* were not identified to be similar. In particular, MLST analysis of patient no. 3 was performed using seven alleles, suggesting that different genotypes of *H. pylori* inhabit the oral cavity and gastric mucosa.

### 2.5. Phylogenetic Analysis

Phylogenetic trees were created using the samples that obtained a combination of two or more alleles (Figure 2). There was a large sequence of diversity between strains of oral and gastric origin in most samples. On the other hand, the oral-derived *H. pylori* strains in patients no. 3, 6, 12, 15, 17 and 21 had markedly similar sequences to their respective counterpart gastric-derived strains, revealing that evolution distances were 0.03 or less, with bootstrap values of 100. These data imply the possibility that oral and gastric *H. pylori* are related in a small number of patients. Partial sequences per patient are demonstrated in the Appendix A.

## 3. Discussion

In the present study, we compared the genome sequences of gastric and oral *H. pylori* from 21 patients with early gastrointestinal cancer using MLST. MLST analysis is an unambiguous procedure that characterizes isolates of bacterial species using the sequences of internal fragments of housekeeping genes. MLST analysis can also help predict the history of human migrations [25]. To our knowledge, this is the first report of the molecular epidemiological analysis of the homology between gastric and oral *H. pylori*.

We were able to obtain *H. pylori* DNA from both the stomach and the oral cavity of all patients who tested positive through a fecal antigen test. These findings supported previous epidemiological studies showing that *H. pylori* colonization in the oral cavity is associated with its presence in gastric tissue [18,32].

Oral *H. pylori* DNA was frequently detected in the supragingival biofilm on the upper incisor teeth (Table 2). The characteristic distribution may be due to local oxygen concentration and environmental acidity. Since anterior teeth are frequently exposed to oxygen, the local oxygen concentration may create a favorable environment for the bacteria. Supragingival biofilm may also provide an optimum environment for *H. pylori* due to its acidic-maintaining property.

In almost all the phylogenetic trees obtained using the seven housekeeping genes, *H. pylori* sequences in this study classified close to the Asian strain that is considered to be high pathogenic (Appendix A). Some *H. pylori* strains involved in carcinogenesis have cytotoxin-associated genes such as *cagA* and *vacA* [33,34]. An isolate from East Asia often carries these genes with unique sequences, which is associated with enhanced virulence [35]. Yamaoka et al. reported that the carrying rate of the highly pathogenic *vacA* gene was significantly higher in East Asian strains (94.7%) than that in non-Asian strains (0.1%) [36]. Conversely, bacteria with weak expression of these genes are classified as attenuated or non-toxic strains [37].

The findings obtained from the MLST analysis were extremely limited. The number of *H. pylori* allelic profiles of eight loci obtained from oral samples ranged from zero to eight. One of the reasons for the failure to acquire the allelic profile is that the yield of DNA was small even when the nested PCR was performed. At present, no reliable isolation culture method for oral *H. pylori* has been established. Therefore, researchers are using the nested PCR method to detect the very rare oral *H. pylori*. However, the sensitivity of the nested PCR varies according to the primer set. For instance, the nested PCR of the combinations of EHC/ET-5 is incapable of detecting *H. pylori* unless oral *H. pylori* exist at concentrations exceeding one in a million [13]. Sulo et al. reported that the abundance and threshold of *H. pylori* present in the oral cavity are low, and the possibility of DNA breakage is high, so the sensitivity of primers with long bp is considerably reduced [38]. Most of the primers targeting the house-keeping DNA used in this study have more than 500 bp in the final amplification product, which is longer than the 228 bp of EHC/ET-5. To overcome this limitation, scholars need to establish an isolation culture method that enables genetic heterogeneity analysis, such as through whole-genome sequencing [39], Core Genome MLST [40], and pulse-field gel electrophoresis [41].

MLST results showed that one out of the 21 patients (patient no. 13) harbored the same genotype, indicating that the strains in the stomach and oral cavity may be of the same origin (Table 4). However, the specificity was low because only two housekeeping genes were used for the analysis. Moreover, although MLST was possible with one gene, the number of candidate MLST numbers increased accordingly, and the specificity and reliability were low due to the fact that six or more alleles are required to generate reliable results [42]. MLST analysis of patient no. 3, which was performed using seven alleles, showed the different candidates between gastric and oral *H. pylori* (Table 4). This result indicated with high confidence that the oral and gastric *H. pylori* were of different origin.

We created a phylogenetic tree using the samples that obtained a combination of two or more alleles, including *vacA* (Figure 2). The phylogenetic analysis of patient no. 3 was performed using the sequences of eight genes. The result showed that the oral *H. pylori* strains were closely related to the corresponding gastric *H. pylori*. In addition, the analyses of patient no. 6, 12, 15, 17 and 21 were performed using the sequences of several genes (no. 6; *mutY* and *trpC*, no. 12; *ppa* and *yphC*, no. 15; *efp*, *mutY* and *vacA*, no. 17; *Urel*, *mutY* and *yphC*, no. 21; *Urel* and *yphC*), and those two strains were closely related. Thus, gastric and oral *H. pylori* in some patients are genetically similar but not of the same origin, and those strains may be infected on separate occasions. Although the analyses of other patients were also performed using only a few sequences, the results suggested that the oral and gastric *H. pylori* were likely of completely different origins. Another genotype of *H. Pylori* may have infected the oral cavity after the initial infection of the stomach. Since the patients with an *H. pylori* infection in the stomach continue to live in an environment where *H. pylori* are abundant, they may have more opportunities for *H. pylori* re-infection.

Our findings are consistent with a study by Wongphutorn et al. The genotypes of *H. pylori* from saliva and stool samples were compared using a partial *vacA* gene sequence. For seven out of 12 individuals, saliva and stool sequences fell into different clusters on a phylogenetic tree, indicating intra-host genetic variation of *H. pylori*. Although this study used only one gene for their comparison, nearly half of the pairs had different genotypes [43].

However, the identity of gastric and oral *H. pylori* cannot be completely denied because of the following two possibilities. One possibility is a genetical mutation due to the environmental difference. The oral cavity is a challenging environment for microbial survival, since it undergoes high daily fluctuations in nutrient supply, temperature, pH, shear and mechanical forces from mastication and hygiene practices, and chemical exposure such as hygiene and pharmaceutical products [44]. Although the two related strains of six patients were originally of the same origin, the environmental difference may cause a mutation in the housekeeping genes, resulting in different allelic profiles. Although housekeeping genes are highly conserved [45,46], studies have reported that some housekeeping genes can mutate in response to changes in a specific environment [47,48]. Linz et al. reported that the mutation rate during the acute phase of *H. pylori* infection is more than 10 times faster than during the chronic infection phase [49]. Based on this, it is possible that *H. pylori* which originally invaded the oral cavity may have acquired some gene mutations during infection of the stomach, hence the discrepancy in the strain types.

Another possibility is mixed infections of *H. pylori* in the oral cavity and the stomach. Although the two *H. pylori* analyzed in this study were genetically mismatched, the same strain may be present in both the oral cavity and the stomach. There are some studies reporting mixed infections of multiple types of *H. pylori* in the oral cavity and stomach [50,51,52]. Palau et al. reported that, based on housekeeping genes, different strains of *H. pylori* were detected from the same site in the stomach of the same patient. This may also occur in the oral cavity [52]. Thus, a future study comparing *H. pylori* DNA extracted from samples taken at different locations is necessary.

In contrast, there were some studies reporting the homology of *H. pylori* in stomach and oral cavity [35,53,54]. Wang et al. compared *cagA* and *vacA* genotypes of *H. pylori* strains from both saliva and stomach in 31 patients with gastritis and peptic ulcer by PCR. The gastric sample was collected via biopsy from the antrum. The results showed 95% agreement between stomach *H. pylori* isolates and their corresponding saliva DNA in at least one cytotoxin genotype. The authors concluded that the same *H. pylori* strain may exist in the saliva and stomach in the same patient. However, the concordance rate of all four cytotoxin genotypes was only 27%, indicating considerable diversity between two comparison targets. Although DNA sequencing from three patients showed 66.9% to 78% homology of *H. pylori* from both sources, data for the other 28 subjects were not provided [55]. Since most studies that analyzed the homology of *H. pylori* in the stomach and oral cavity compared the genotypes of a limited number of pathogenic genes, it is not possible to draw conclusions about the presence or absence of homology in these strains.

Analyzing whether oral and gastric *H. pylori* are genetically identical is important for future strategies of *H. pylori* control. If the genetic relatedness between oral and gastric *H. pylori* was confirmed, the trigger of infection is either the mouth and the stomach at the same time, or the oral cavity functions as a reservoir for *H. pylori*. If *H. pylori* exist alive in the oral cavity and serve as a supply source for the stomach, eradication therapy for *H. pylori* will need to be performed simultaneously in the oral cavity and the stomach. It has been reported that the eradication efficiency in the stomach was 85.8%, while in the oral cavity, it was only 5.7%, revealing 55.6 of the pooled odds ratio (OR) [56].

If a different genotype of *H. pylori* exists in the oral cavity independently of *H. pylori* present in the stomach, the presence of oral *H. pylori* is not a risk factor for gastric infection. Oral *H. pylori* may exist as a part of the normal microflora in the adult oral cavity [13].

Several clinical studies suggested that periodontal therapy combined with *H. pylori* eradication treatment increased the eradication rate of gastric *H. pylori* compared with eradication treatment alone (OR 2.15; 95% Cl 1.47 to 3.14). In addition, the non-recurrence rate of gastric *H. pylori* infection increased in participants treated with periodontal therapy compared with those who received eradication therapy alone (OR 3.60; 95% Cl 2.11 to 6.15) [22]. Although these studies support the claim that the oral cavity is an important reservoir for gastric *H. pylori* infection, it has not been proven that *H. pylori* exist alive in the oral cavity, nor that a species genetically identical to oral *H. pylori* was detected in the stomach after the eradication. Within the limitations of this study, the methodology for analyzing the genetic relationship between oral and gastric *H. pylori* using MLST may be useful for discussing the necessity of eradication of oral *H. pylori.*

## 4. Materials and Methods

### 4.1. Participants

The study was conducted at Niigata University Medical and Dental Hospital (Niigata, Japan) between November 2019 and October 2020. We recruited patients, with a confirmed *H. pylori* infection using a fecal antigen test, who planned to admit for endoscopic surgery on the upper gastrointestinal cancers. The presence of at least one tooth in the oral cavity was part of the inclusion criteria, and participants using dentures were not excluded. The study protocol was approved by the Niigata University Ethics Committee (approval number 2019-0220), and the study was carried out in accordance with the approved guidelines. All participants signed an informed consent form before participating in the study.

### 4.2. Oral Sample Collection

An oral examination, followed by a collection of saliva and biofilm on teeth and tongue, was performed prior to endoscopic surgery. Smoking and oral hygiene status, such as frequency of oral cleaning and use of oral cleaning aids, were recorded. The number of decayed, missing, and filled teeth (DMFT) was also recorded. Unstimulated saliva (2 mL) was collected by spitting into a tube. Supragingival dental biofilm samples were collected by scraping from the upper incisors, lower incisors, upper right molars, and lower left molars using a sterile curette. If the participant had no teeth in the designated location, the sample was taken from the opposite site or denture. Each sample was transferred into a tube containing phosphate-buffered saline (PBS; pH 7). The superficial layers of the tongue were collected using five gentle strokes from the papillae circumvallatae to the anterior part of the tongue dorsum with a tongue brush (Tongue Cleaner Plus, Ci Medical, Ishikawa, Japan). Bacterial cells were retrieved by vigorous stirring in 20 mL PBS. The samples were centrifuged for 10 min at 10,000 rpm, washed twice with PBS, and stored at −80 °C until further use.

### 4.3. Gastrointestinal Endoscopy and Histologic Examination

Biopsy specimens were taken from three locations in the stomach using an endoscope: one from the vestibular region and the other two from the body of the stomach. A part of the samples was fixed with 10% formaldehyde and histologically examined by both Giemsa and H-E staining. The pieces were mixed and crushed with a Power masher II (Nippi, Incorporated, Tokyo, Japan) immediately after collection, and half of the amount was inoculated onto *Helicobacter* Selective Agar medium (Nissui Pharmaceutical co., Ltd., Tokyo, Japan) and incubated at 37 °C under slightly aerobic conditions (5% O_2_, 10% CO_2_, and 85% N_2_), for 72 h. The other half was stored at −80 °C until further use.

### 4.4. DNA Extraction and Nested PCR

DNA was extracted from the oral samples and biopsy specimens in the stomach as well as from a control strain, *H. pylori* NCTC11637, using the NucleoSpin^®^ Microbial DNA Kit (TaKaRa Bio, Shiga, Japan) according to the manufacturer’s instructions. The quantity and purity of the DNA were assessed by spectrophotometry at 260/280 nm. The DNA was stored at −80 °C until processing.

An aliquot of the DNA extract was amplified using nested PCR as previously described [13], and the presence of *H. pylori* DNA in the sample was confirmed by 1.5% agarose gel electrophoresis. Briefly, the first round of PCR amplification was performed with Takara Ex Taq^®^ Hot-Start Version (RR006A; TaKaRa Bio, Shiga, Japan) using the EHC-U (5′-CCCTCACGCCATCAGTCCCAAAAA-3′) and EHC-L (5′-AAGAAGTCAAAAACGCCCCAAAAC-3′) primers targeting an 860-bp fragment of *H. pylori* genomic DNA. The amplification comprised 40 cycles of denaturation at 98 °C for 10 s, annealing at 57 °C for 30 s, and extension at 72 °C for 1 min on a MiniAmp^TM^ Thermal Cycler (Thermo Fisher Scientific, Waltham, MA, USA). The expected product size was 417 bp, covering the area from 80,076 bp to 80,492 bp in the *H. pylori* genome [10].

For the nested PCR assay, the amplification product (1 µL) obtained by single-step PCR was re-amplified over 20 cycles under the same conditions as in the first round. The internal primer pair ET-5U (5′-GGCAAATCATAAGTCCGCAGAA-3′) and ET-5L (5′- TGAGACTTTCCTAGAAGCGGTGTT-3′) was used at a concentration of 50 pmol/µL. The expected size was 228 bp. A sample, in which the presence of *H. pylori* DNA was confirmed, was used for the subsequent analysis.

### 4.5. Sequencing and MLST Analysis

MLST was performed as described on PubMLST (http://pubmlst.org/helicobacter/, accessed on 8 January 2023) [30]. Primer sets targeting seven housekeeping genes, *atpA*, *ureI*, *efp*, *mutY*, *ppa*, *trpC*, and *yphC* were used. The sequences were designed with reference to the MLST website or an article by Osaki et al. (Table 5). Gene fragments containing these genes were amplified from *H. pylori*-positive specimens by the nested PCR as described above.

Briefly, the first round of PCR amplification was performed using Takara Ex Taq^®^ Hot-Start Version. The amplification comprised 40 cycles of denaturation at 98 °C for 10 s, annealing at 57 °C of each primer for 30 s, and extension at 72 °C for 1 min on a MiniAmp™ Thermal Cycler. For the nested PCR assay, the amplification product (1 µL) obtained by single-step PCR was re-amplified over 40 cycles under the same conditions as in the first round. *H. pylori* NCTC11637 DNA served as the positive control, and water was used as the negative control. Each PCR product was confirmed by 1.5% agarose gel electrophoresis.

The band visualized under LED light was cut out from the gel using a gel band cutter (FastGene ™ Agarose Gel Band Cutter, Nippon Genetics co., Ltd., Tokyo, Japan). Where multiple bands were present, bands of the same size as the positive control band were collected. Amplicons were purified from the gel slices using the Freeze’N squeeze DNA Gel Extraction Spin Clumns (Bio-Rad, Hercules, CA, USA) and NucleoSpin^®︎^ Gel and PCR Clean-up (TaKaRa Bio, Shiga, Japan). The amplified DNA was then sequenced and analyzed using the Applied Biosystems 3730xl DNA analyzer at Macrogen Japan corp. (Tokyo, Japan).

The sequences were uploaded onto PubMLST (http://pubmlst.org/helicobacter/, accessed on 8 January 2023) to determine the closest allele type for each gene. Using the allelic profile of each gene, the sequence type of the sample was determined.

### 4.6. Phylogenic Tree Analysis

To analyze the phylogeny of *H. pylori* obtained from the oral cavity and stomach, housekeeping gene sequences of 159 *H. pylori* isolates from 29 countries, in eight regions of the world, were downloaded from the BLAST website (https://blast.ncbi.nlm.nih.gov/Blast.cgi, accessed on 8 January 2023) (Appendix A).

Multiple alignments of the MLST genes was then performed using MEGA (V11). The aligned sequences were used to develop a phylogenetic tree. The amplicons obtained from the mouth and stomach were then sequenced and aligned with isolates from Europe, Africa, Asia, America and Oceania using multiple alignments with the MUSCLE program. Asia was classified into East Asia, Southeast Asia, and South Asia. Thereafter, a phylogenetic tree was constructed using the construct/Test Maximum Likelihood tree with the alignment result in MEGA. Unbiased bootstrap values were used to support the tree using MEGA’s Bootstrap Replications. A cluster that was bootstrapped 1000 times and had a bootstrap value of 95 or higher was considered supported.

### 4.7. Statistical Analysis

Data analysis was carried out using SPSS^®^ 11.0 (SPSS, Chicago, IL, USA). The chi-squared test and Fisher’s exact probability test were used when applicable, and the results were considered statistically significant when the *p*-value was <0.05. Prevalence was expressed as a proportion and the OR was used to measure the strength of the association between the variables.

## 5. Conclusions

In conclusion, within the limitations that require careful interpretation, different genotypes of *H. pylori* exist in the oral cavity independently of *H. pylori* present in the stomach in most cases. There are some cases in which nearly related *H. pylori* strains with close evolutionary distance are present in the stomach and the oral cavity. Oral *H. pylori* may exist as a part of the normal microflora in the adult oral cavity.

A method using MLST for analyzing the genetic relationship between oral and gastric *H. pylori* may be useful for scientific elucidation of infection routes and discussing the necessity of eradication of oral *H. pylori.* It is necessary to establish a culture method for oral *H. pylori* to elucidate whether the oral cavity acts as the source of gastric infection, as our analysis was based on a limited number of combinations of allele sequences.

## Figures and Tables

**Figure 1 ijms-24-02211-f001:**
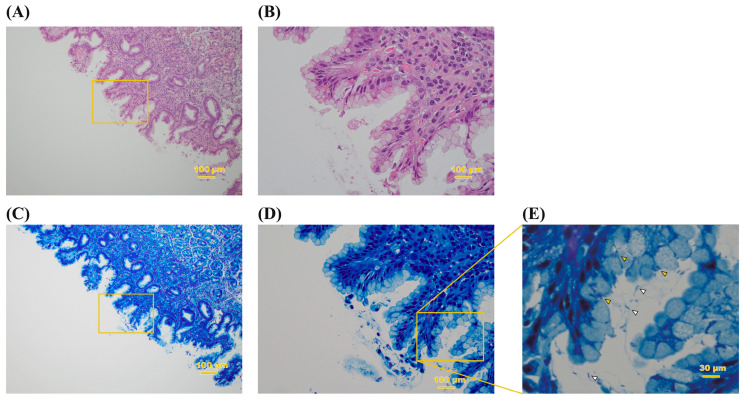
Histopathological observation of biopsy specimen. The gastric mucosa stained with HE (**A**,**B**) and Giemsa (**C**–**E**) of patient no. 15. The original magnification is ×10. (**B**,**D**) Higher magnification of the area indicated by the squares in (**A**,**C**), respectively. (**E**) High-magnification image of the box in (**D**). Spiral-like and seagull-like bacilli can be confirmed in the mucus (white arrowhead) and the interior of mucous cells (yellow arrowhead) of the stomach.

**Figure 2 ijms-24-02211-f002:**
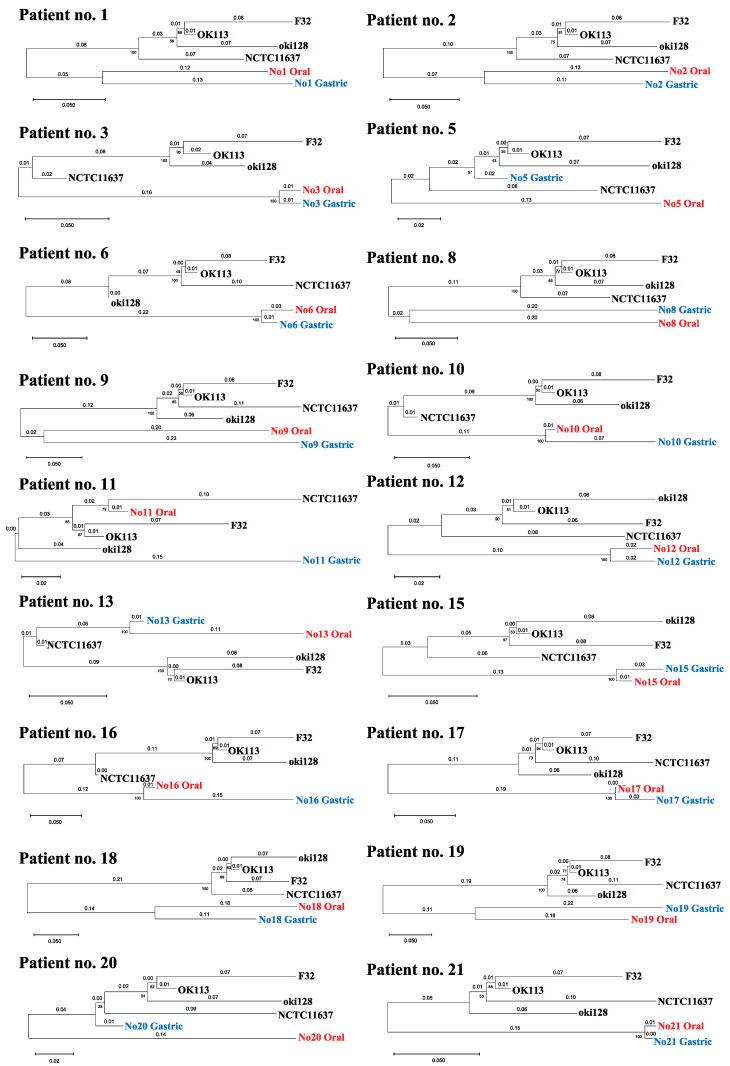
Phylogenetic tree showing the correlation of oral and gastric *H. pylori* based on sequence arrangements. Phylogenetic tree was created using the samples that obtained a combination of two or more alleles, including patient no. 13. The gene sequences of *H. pylori* OK113, oki128, and F32 clinical isolates from East Asia were downloaded from the BLAST website. The numbers indicate distances between the strains tested (substitutions/site).

**Table 1 ijms-24-02211-t001:** Demographic and clinical characteristics of the patients.

Patient No.	Sex	Age	No. of Remaining Teeth	DMFT	Frequency of Oral Cleaning	Cleaning Aid	Smoker	GI Cancer			
								Location	Histological Type	Differentiation	Depth of Tumor Invasion (T-Category)
1	F	78	18	18	1	−	−	Corpus	Adenocarcinoma	tub1 > tub2	T1a
2	F	66	28	11	3	+	−	Corpus	Adenocarcinoma	tub1, tub2	T1a
3	M	66	15	22	2	−	−	Corpus	Adenocarcinoma	tub1	T1b2
4	M	76	24	18	3	−	−	Corpus	Adenocarcinoma	tub1	T1a
5	F	67	8	28	2	+	−	Corpus	Adenocarcinoma	tub1	T1a
6	F	75	29	24	2	+	−	Corpus	Adenocarcinoma	tub1	T1a
7	M	81	13	24	1	+	−	Antrum	Adenocarcinoma	tub1 > tub2	T1a
8	M	63	28	20	2	−	+	Corpus	Adenocarcinoma	tub1	T1a
9	M	86	5	28	1	−	−	Antrum	Adenocarcinoma	tub1	T1a
10	M	59	26	22	1	−	+	Corpus	Adenocarcinoma	tub1	T1a
11	M	78	29	5	1	−	−	Corpus	Adenocarcinoma	tub1 > tub2	T1a
12	F	77	24	13	2	+	−	Corpus	Adenocarcinoma	tub1, tub2	T1a
13	M	81	26	9	2	+	−	Antrum	Adenocarcinoma	tub1	T1a
14	F	84	28	11	2	+	−	Corpus	Adenocarcinoma	tub1	T1a
15	F	70	25	17	2	+	−	Corpus	Adenocarcinoma	Por2, sig	T1a
16	M	66	30	9	2	+	+	Corpus	Adenocarcinoma	tub1, tub2	T1b1
17	M	76	14	21	1	−	−	Corpus	Adenocarcinoma	tub1 > tub2	T1a
18	M	79	15	27	0	−	−	Corpus	Adenocarcinoma	tub1	T1a
19	M	65	28	10	3	+	−	Antrum	Adenocarcinoma	tub1, pap-tub1 > tub2	T1b2
20	M	70	12	25	2	−	−	Abdominal esophagus; Corpus	Squamous cell carcinoma; Adenocarcinoma	Mod; tub1	T1b1
21	M	66	6	26	3	−	+	Corpus	Adenocarcinoma	tub1, low grade	T1a
Mean ± SD		72.8 ± 7.6	20.5 ± 8.4	18.5 ± 7.2	1.8 ± 0.8						

DMFT: Decayed, missing, and filled teeth; GI: gastrointestinal; Tub1: Tubular adenocarcinoma well differentiated type; Tub2: Tubular adenocarcinoma moderately differentiated type; Por2: poorly differentiated adenocarcinoma non-solid type; Sig: Sig-net-ring cell carcinoma; Mod: Moderately differentiated. T1a; Tumors limited to the mucosal layer. T1b1; Tumors with penetration to the submucosal layer less than 500 μm from the muscularis mucosae. T1b2; Tumors with penetration of 500 μm or more from the muscularis mucosae.

**Table 2 ijms-24-02211-t002:** Prevalence of *H. pylori* DNA in gastric tissue and by location in the oral cavity using nested PCR.

Patient No.	Upper Incisor	Lower Incisor	Upper Right Molar	Lower Left Molar	Saliva	Tongue	Gastric Tissue Biopsy	Gastric Tissue Culture
1	+	+	−	−	−	−	+	+
2	+	−	−	−	−	+	+	+
3	−	−	+	−	−	−	+	+
4	+	−	−	−	−	−	+	+
5	+	−	−	−	−	−	+	+
6	−	−	+	−	−	+	+	+
7	−	+	−	−	−	+	+	+
8	+	−	−	−	−	−	+	+
9	+	−	+	−	-	−	+	+
10	+	−	−	−	+	−	+	+
11	−	−	−	−	−	+	+	+
12	+	−	−	−	−	−	+	+
13	+	−	−	−	+	−	+	+
14	+	−	+	−	−	+	+	+
15	+	+	−	−	+	−	+	+
16	+	−	−	−	−	+	+	+
17	+	−	−	+	−	−	+	+
18	+	−	−	−	−	−	+	+
19	+	−	+	−	−	−	+	+
20	−	+	−	−	−	−	+	+
21	−	−	−	−	+	−	+	+
Prevalence (%)	71.4 *	19	23.8	4.8	19	28.6	100	100
*n/N*	15/21	4/21	5/21	1/21	4/21	6/21	21/21	21/21
OR	10.6	1	1.32	0.21	1	1.7		

* *p* < 0.05, compared with other sites.

**Table 3 ijms-24-02211-t003:** Allele analysis of *H. pylori* DNA sequences obtained from oral and gastric samples targeting eight housekeeping genes.

Participants No.	Housekeeping Gene															
	*atpA*		*urel*		*efp*		*mutY*		*ppa*		*trpC*		*yphC*		*vacA*	
	Oral	Gastric	Oral	Gastric	Oral	Gastric	Oral	Gastric	Oral	Gastric	Oral	Gastric	Oral	Gastric	Oral	Gastric
1	-	NP	n/m	3242	-	NP	-	NP	-	NP	-	NP	3504	3488	-	NP
2	-	NP	n/m	463	453	n/m	-	NP	-	NP	-	NP	-	NP	-	NP
3	2364	963	2643	2943	1162	2421	2368	2368	45	942	458	970	3556	3538	573	573
4	-	NP	-	NP	-	NP	-	NP	-	NP	-	NP	-	NP	-	NP
5	-	NP	-	NP	-	NP	-	NP	n/m	823	-	NP	3572	3542	-	NP
6	-	NP	-	NP	-	NP	n/m	2476	-	NP	970	423	-	NP	-	NP
7	-	NP	n/m	2647	-	NP	-	NP	-	NP	-	NP	-	NP	-	NP
8	-	NP	n/m	67	-	NP	-	NP	942	929	-	NP	-	NP	-	NP
9	-	NP	286	2856	-	NP	-	NP	849	45	-	NP	-	NP	-	NP
10	2364	1760	n/m	466	-	NP	-	NP	-	NP	-	NP	3513	457	-	NP
11	-	NP	-	NP	-	NP	n/m	2360	420	945	-	NP	3569	3515	-	NP
12	-	NP	-	NP	-	NP	-	NP	420	448	-	NP	3515	1962	-	NP
13	2003	958	-	NP	-	NP	n/m	2729	-	NP	-	NP	-	NP	-	NP
14	-	NP	-	NP	-	NP	935	38	-	NP	-	NP	-	NP	-	NP
15	-	NP	-	NP	453	1879	935	1907	-	NP	-	NP	-	NP	29	950
16	1760	n/m	3243	966	-	NP	-	NP	-	NP	-	NP	3538	3523	-	NP
17	-	NP	n/m	2471	-	NP	935	2450	-	NP	-	NP	3572	3572	-	NP
18	-	NP	n/m	466	-	NP	3162	935	-	NP	-	NP	-	NP	-	NP
19	-	NP	n/m	825	-	NP	935	1219	-	NP	-	NP	-	NP	785	618
20	-	NP	-	NP	-	NP	-	NP	1887	445	-	NP	3513	3543	-	NP
21	-	NP	738	971	-	NP	-	NP	-	NP	-	NP	3542	3542	-	NP

Nested PCR using primers targeting the housekeeping genes did not give amplification products with all the primers. Allele analysis was performed on the obtained sequences to determine the allele sequence types. Some of them were new sequence types that were unclassified (no matching (n/m)). NP: not performed.

**Table 4 ijms-24-02211-t004:** MLST profiles of oral and gastric *H. pylori*.

Patient No.	Specimen Type	Candidate MLST
1	Oral	n/m
	Gastric tissue culture	4001, 4023
2	Oral	1160, 1473, 2830
	Gastric tissue culture	669, 1290
3	Oral	2843
	Gastric tissue culture	1173, 1324, 1441, 2749, 2784, 2806, 2809, 2810, 2832, 2848, 3032, 3299, 3340, 3427, 3590, 3720, 3724, 3725, 3742
5	Oral	n/m
	Gastric tissue culture	1502, 1741, 1836, 3572.
6	Oral	1173
	Gastric tissue culture	550, 2822, 2830, 3034, 3442, 3666
7	Oral	n/m
	Gastric tissue culture	3035
8	Oral	1324, 2749, 2806, 2832, 2848
	Gastric tissue culture	425, 660
9	Oral	286, 1869
	Gastric tissue culture	45, 2836, 3302, 3339
10	Oral	2843
	Gastric tissue culture	2753, 2754, 2755, 2810
11	Oral	550, 2822, 2835, 2847
	Gastric tissue culture	1228, 2750, 2756, 2757, 2758, 2790, 2807, 2827
12	Oral	550, 854, 2822, 2835, 2847
	Gastric tissue culture	52, 1064, 1154, 1661, 2081, 2115, 2218, 2257, 2835, 3325, 3333, 3348, 3371, 3453, 3459, 3700, 3701, 3702, 3722, 3733, 3734, 3997, 4003, 4051, 4070, 4083
13	Oral	2207, **3351**
	Gastric tissue culture	1228, 3206, 3331, 3347, **3351**, 3352, 3370, 3376, 3429, 3436, 3464, 3466
14	Oral	854
	Gastric tissue culture	38, 42
15	Oral	854, 1160, 1473, 2830
	Gastric tissue culture	2123, 2258, 3332, 3335, 3414, 3422, 3436, 3442, 3478
16	Oral	2226, 2753, 2754, 2755, 2816, 2817, 2841, 4001, 4023
	Gastric tissue culture	1324, 1474, 2795, 3660, 3995
17	Oral	854
	Gastric tissue culture	2802, 2811, 2812
18	Oral	3856
	Gastric tissue culture	854, 1160, 3328
19	Oral	854
	Gastric tissue culture	770, 921, 3350, 3405, 3446
20	Oral	n/m
	Gastric tissue culture	1473, 2761, 2799, 2816, 2830, 2841
21	Oral	1349
	Gastric tissue culture	1042

Sample sets with matching sequence types are shown **in bold.** MLST: Multilocus sequence typing; n/m: no matching.

**Table 5 ijms-24-02211-t005:** Primers for amplification of housekeeping genes using nested PCR.

Locus	PCR	Name	Primer *	Amplicon (bp)	Annealing (°C)	Reference
*atpA*	First	atpA_for2	GGACTAGCGTTAAACGCACG	840	57	MLST website
		atpA_rev2	CTTGAAACCGACAAGCCCAC			MLST website
	Second	atpA_for3	GTTCCTGTTGGCGATGCGGT	766	57	MLST website
		atpA_rev3	CCTGAATAAAACAAATCCGTTTC			MLST website
*urel*	First, second	urelfor	AGGTTATTCGTAAGGTGCG			MLST website
	First	urel-rev3	GAAATCCAAGGGGTTTAAATC	695	57	MLST website
	Second	urel-rev2	GTTTAAATCCCTTAGATTGCC	683	57	MLST website
*efp*	First	efp_for1	GGCAATTTGGATGAGCGAGCTC	558	57	MLST website
		efp_rev1	CTTCACCTTTTCAAGATACTC			MLST website
	Second	efp_for2	GGGCTTGAAAATTGAATTGGGCGG	500	57	MLST website
		efp_rev2	GTATTGACTTTAATGATCTCACCC			MLST website
*mutY*	First	mutY_for4	TTATGAAGTCTCTATATCAGCGAAGT	529	54	MLST website
		mutY_rev4	TACCTAAACAATAAGGATTGAAAGG			MLST website
	Second	mutY_for5	ATATCAGYGAAGTGATGAGC	516	50	MLST website
		mutY_rev5	CCYAAACAATAAGGRTTKGAA			MLST website
*ppa*	First	ppa_for1-1	GAARTKAGCCATGACGCTRA	698	54	MLST website
		ppa_rev4	GGGTTAARATCGTTAAATTGTAG			MLST website
	Second	ppa_for1-2	AGCCATGACGCTRAKYCTTT	490	54	Osaki et al. (2013) [25]
		ppa_rev1-2	CTCTTTGTTTTCAAACCCCTTG			Osaki et al. (2013) [25]
*trpC*	First	trpC_for8	AGCATCGCCCTCTAAAGGTT	618	57	Osaki et al. (2013) [25]
		trpC_rev6	AAGCCCGCACACTTTATTTTC			Osaki et al. (2013) [25]
	Second	trpC_for9	TCGCCCTCYAAAGGTTTRAT	564	57	Osaki et al. (2013) [25]
		trpC_rev9	TCAAATCCTTTTCTTTCATYA			Osaki et al. (2013) [25]
*yphC*	First	yphC_for2	CACGCCTATTTTTTTGACTAAAAAC	734	54	MLST website
		yphC_rev3	CATTYACCCTCCCAATGATGC			MLST website
	Scond	yphC-for3	GACCCTTATTTAAGCTTTAAATAAC	688	54	MLST website
		yphC-rev3	CCCAATGATGCCTACTTGAAT			MLST website
*vacA*	First	HPVacA1-4	ATACGCTCCCACGTATTGC	695	57	Osaki et al. (2013) [25]
		OLHPVacA-3(+)	ACAACCGTGATCATTCCAGC			Osaki et al. (2013) [25]
	Second	VacA-for2	CTGCTGTAGGAACGGTCTC	575	57	Osaki et al. (2013) [25]
		VacA-rev2	GCGTGGCGCCATCATAAAGAG			Osaki et al. (2013) [25]

* Y = C or T, K = G or T, R = A or G.

## Data Availability

The datasets presented in this study can be found in online repositories. The names of the repository/repositories and accession number(s) can be found below DDBJ. Accession numbers are shown in the article/Appendix A.

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
