# Peer review of "Analysis of Genetic Relatedness between Gastric and Oral Helicobacter pylori in Patients with Early Gastric Cancer Using Multilocus Sequence Typing"

_ijms, 2023, doi:10.3390/ijms24032211_

Round 1

Reviewer 1 Report

In this study, Ryoko Nagata et al reported genetic relatedness between gastric and oral H. pylori in patients with early gastric cancer using multilocus sequence typing. They found different genotypes of H. pylori inhabit the oral cavity and gastric mucosa. The phylogenetic analysis showed oral and gastric

strains were related but not of the same origin, and those strains may be infected on separate occasions. It is an interesting study. However, I have some questions:

1.      In the topic, early gastric cancer was mentioned. In fact, one esophageal squamous cell carcinoma was included in this study. Do authors mean a patient with double cancers (gastric adenocarcinoma and esophageal SCC)? And what is the definition of early gastric cancer in this study? According to the previous studies, patients with gastric antral/body adenocarcinoma are more related to H. pylori infection than those with gastric cardiac adenocarcinoma. Authors should clarify the locations (antrum, body or cardiac) of gastric adenocarcinoma. They may perform a subgroup analysis only including patients with gastric antra/body adenocarcinoma.

2.      The environments of oral cavity and stomach are quite different (pH value, light exposure, oxygen concentration…). It is reasonable to find different strains of H. pylori between oral cavity and stomach, for adopting the different environments. In line 214-217, authors mention "gastric and oral H. pylori in some patients are genetically similar but not of the same origin, and those strains may be infected on separate occasions. These findings also supported that oral H. pylori is unlikely to be colonized by acid reflux”. Because the past study by Wang et al. using samples from saliva and gastric antral tissue from patients with peptic ulcer found high (95%) agreement between stomach H. pylori isolates and their corresponding saliva DNA in at least one cytotoxin genotype. Gastroesophageal reflux content in the saliva could not be excluded out. Similarly, reflux content in the oral cavity may still happen in patients with multiple H. pylori strains in the stomach.

3.      Could authors explain the selection of housekeeping genes for MLST in this study? Why vacA not cagA gene selected, adding on the other seven genes (afpA, afp, mufY, ppa, trpC, urel, ypbc)?

4.      Line 226, “Therefore, we speculate that both gastric and oral H. pylori detected in this study are virulent”. Could author explain more detail how to get this speculation from this study?

Reviewer 2 Report

1. The study aims to evaluate the origins of H. pylori strains in oral and gastric tissues by sequencing serval conserved loci of the bacterial gene. While the methodology was clear and straightforward, the authors should provide more context regarding the significance and meaning of such work. In other words, what could we gain from understanding if the H. pylori are or are not from the same origin in oral and gastric tissues? Will this knowledge help us with disease prevention, diagnosis, or treatment at any level? One would imagine that the same or different strains of H. pylori could exist in oral and gastric tissues, and it’s even possible different strains could co-exist in the same tissue location. The authors should provide more rationale for this particular study rather than simply suggesting that such work has not been performed yet. 

2. The extraction of H. pylori DNA was not so successful from many patient samples. Was the DNA extracted directly from the tissue samples? It seems bacteria colonies were cultured on agar plates and were used to extract DNA in SIFig1. Have the authors tried to extract DNA from bacterial cultures to improve yield? If not, why? 

3. In Fig.1, what information do the authors trying to deliver with representative images from a single patient sample? The tissue of gastric cancer, or tissue with H. pylori infection? Whatever the authors try to highlight, it’s better to include a negative control for comparison (e.g same images from normal gastric tissue). If normal tissue control is not possible, then images from at least multiple patients should be shown. 

4. The color of the text in Fig.2 is very hard to differentiate. Actually, the font/style of the texts is different in the figure panels. This figure needs to be revised to make it more readable and clear. 

Reviewer 3 Report

The authors analyzed the genetic relatedness of H. pylori in gastric and oral sites of early gastric cancer patients with multilocus sequence typing. The introduction and methods are well written. The conclusion of this work is not clear and the contribution to the field is little.

Round 2

Reviewer 2 Report

The main concerns were properly addressed. 

Reviewer 3 Report

All points has been addressed.